# Fast volumetric fluorescence lifetime imaging of multicellular systems using single-objective light-sheet microscopy

Valentin Dunsing-Eichenauer [1,2,8] ✉, Johan Hummert [3,8] ✉, Claire Chardès [1], Thomas Schönau[3], Léo Guignard [1], Rémi Galland[4], Gianluca Grenci [5,6], Max Tillmann[3], Felix Koberling[3], Corinna Nock[3], Jean-Baptiste Sibarita [4], Virgile Viasnoff[5], Ivan Michel Antolovic[7], Rainer Erdmann[3] & Pierre-François Lenne [1]

Fluorescence lifetime imaging (FLIM) is widely used for functional and multiplexed bioimaging. The lifetime of autofluorescence or fluorescent sensors encodes physiologically relevant parameters. Thus, FLIM is especially relevant for the investigation of living systems. However, application of FLIM to live specimen is hampered by its slow speed and high phototoxicity. To enable faster and gentler FLIM, we integrated single-objective light-sheet microscopy with pulsed excitation and time-resolved detection on a novel SPAD array detector. We achieved 10–100-fold acceleration compared to confocal FLIM, down to 100 ms acquisition time per image, with excellent quantitative agreement. The massively enhanced speed enables volumetric FLIM acquisitions on live multicellular specimens, which we demonstrate with lifetime-based multiplexing in 3D and time-lapse FLIM of tension probes on living embryonic organoids. We benchmark both scanned and static light-sheet modalities to facilitate adding FLIM capability to a large variety of light-sheet microscopes.

Fluorescence lifetime imaging (FLIM) is a powerful technique that enhances the information content of microscopy images beyond fluorescence intensity. Lifetime information can be utilized to multiplex imaging of different targets[1–3]. Lifetime can also encode biophysically relevant parameters such as pH, temperature, membrane tension, and metabolic activity[4–7]. These sensing capabilities make FLIM especially suited for studies on living cells and multicellular systems. However, in its typical implementation in confocal laser scanning microscopes, the applicability of FLIM to live samples is often limited by the trade-off between speed and phototoxicity. Light-sheet microscopy excels in exactly this trade-off, allowing for fast and gentle imaging.

Until recently, combining light-sheet microscopy and FLIM has been hampered by the availability of nanosecond time-resolved detectors for widefield microscopy. Previously employed detectors for light-sheet FLIM suffer from high dark currents in the case of intensified or modulated cameras[8,9], from limited throughput in the case of multichannel plate-based detectors[10], or limited fill factor and sensor size in the case of novel SPAD arrays[11–13]. Promising approaches for high-speed FLIM of live samples are

time-gating via fast Pockels cells[14,15] or compressed sensing[16], but both require specialized technology that is not readily available. Therefore, light-sheet FLIM has not yet realized its potential to enable faster lifetime imaging than state-of-the-art laser scanning microscopes. In addition, no quantitative comparison of light-sheet and confocal FLIM going beyond analysis of average lifetime has been performed.

Here, we overcome these limitations by integrating single-objective light-sheet microscopy[17,18] (soSPIM) with a newly available SPAD array detector. The soSPIM technology utilizes 45° mirrors embedded in microfabricated imaging devices that create a light-sheet perpendicular to the optical axis and collect the excited fluorescence through the same high numerical aperture (NA) objective used for illumination. Benefitting from this, high-resolution imaging, super-resolution single molecule localization microscopy[17], and imaging fluorescence correlation spectroscopy using soSPIM[18] have been demonstrated in model organisms and culture cells. Recently, it has also been shown that soSPIM facilitates high-content 3D imaging of a variety of organoid specimens (e.g., human gastruloids, intestinal, hepato- and neuroectoderm organoids, oncospheres)[19,20], thanks

[1]Aix-Marseille Université, CNRS, IBDM - UMR7288, Turing Centre for Living Systems, Marseille, France. [2]Department of Infectious Diseases and Respiratory Medicine, Charité-Universitätsmedizin Berlin, Berlin, Germany. [3]PicoQuant GmbH, Berlin, Germany. [4]Univ. Bordeaux, CNRS, Interdisciplinary Institute for Neuroscience, IINS, UMR 5297, Bordeaux, France. [5]Mechanobiology Institute, National University of Singapore, Singapore, Singapore. [6]Biomedical Engineering Department, National University of Singapore, Singapore, Singapore. [7]Pi Imaging Technology SA, EPFL Innovation Park, Lausanne, Switzerland. [8]These authors contributed equally: Valentin Dunsing-Eichenauer, Johan Hummert. ✉e-mail: valentin.dunsing-eichenauer@charite.de; hummert@picoquant.com

to dedicated cell culture chips containing arrays of wells, each flanked with four micromirrors (JeWells).

The high collection efficiency of soSPIM enabled by the use of a high NA objective is particularly advantageous for lifetime imaging, significantly improving upon previous light-sheet FLIM implementations with strongly limited collection efficiency[9,10,12,13], except the work by Nedbal et al. using a specialized tilt illumination device compatible with high NA detection[11]. With soSPIM-FLIM, we achieve FLIM at unprecedented speed in 3D, enabling multiplexing by lifetime and tension sensing in live organoids.

## Results

To extend soSPIM to FLIM, we have equipped a soSPIM microscope with a newly developed pulsed laser and a state-of-the-art SPAD array detector. The laser is a prototype (PicoQuant GmbH) based on fiber amplification of mid-infrared laser diodes and frequency conversion[21]. The detector is a 512 × 512 pixel gated SPAD array (SPAD512[2], Pi Imaging Technology), enabling time-resolved imaging with a nanosecond global exposure gate that can be shifted with picosecond precision relative to the laser pulse (Fig. 1a).

As a benchmark, we performed soSPIM and confocal FLIM measurements on the same samples, AF488 dye solution and fixed embryonic organoids (gastruloids) immunostained with E-cadherin-AF488. Gastruloids are aggregates of mouse embryonic stems cells (see "Methods"), mimicking early embryonic patterning and axis formation[22]. Besides their biological relevance, they represent key imaging challenges in the organoid field as they are sensitive, highly dynamic, and opaque specimen[23]. We evaluated soSPIM-FLIM performance in two light-sheet modalities (Fig. 1a): a digital scanned light-sheet generated by fast beam scanning along micromirrors and a static light-sheet created by a cylindrical lens. For confocal measurements, we used a conventional time-correlated single

photon counting (TCSPC) system limited to ca. 4 MHz photon count rate and a state-of-the-art rapidFLIM system that supports photon count rates of up to 80 MHz[24]. As a metric for FLIM performance, we quantified the average and interquartile ranges of lifetime histograms and photon counts in the brightest pixels, with FLIM frame acquisition times ranging from 100 ms to 1 s for light-sheet and 1 s to 2 min for confocal FLIM (Fig. 1b). For organoid measurements, only pixels within the organoid and above a global intensity threshold were considered in the comparison to exclude background pixels. We found excellent agreement between pixel lifetimes measured by confocal and soSPIM-FLIM (Figs. 1c, and S1–3). However, soSPIM measurements provided a far more precise estimate within much shorter acquisition times. Even in comparison to rapidFLIM, ca. 2× narrower lifetime histograms were achieved in 2× shorter acquisition time in scanning light-sheet mode. In static mode, ~4× narrower histograms were achieved in 10× shorter acquisition time than in rapidFLIM. Correspondingly, ~1000 counts in the brightest pixel and similar histogram widths were achieved in 100 ms with static light-sheet, in 1.1 s with scanning light-sheet, and in 10.9 s with RapidFLIM (Fig. 1c). Strikingly, in comparison to rapidFLIM, high photon counts were generated with 55 and 220 times lower laser power densities for scanning and static soSPIM-FLIM, respectively (Table S1). The parallelisation advantage in using a camera detector is also evident in the count rates. While pixel count rates are in the kHz range in the light-sheet and in the MHz range in the confocal modalities, overall count rates are below 1 MHz for all confocal modalities and ~40 and 500 MHz for the scanning and static light-sheet respectively (Fig. 1c and Table S1).

The shortest acquisition times are achievable only in static light-sheet measurements as the speed of scanning light-sheet FLIM is limited by the available beam scan rates and the potential for pile-up on the SPAD array detector. The scanning light-sheet is much more prone to pile-up due to the

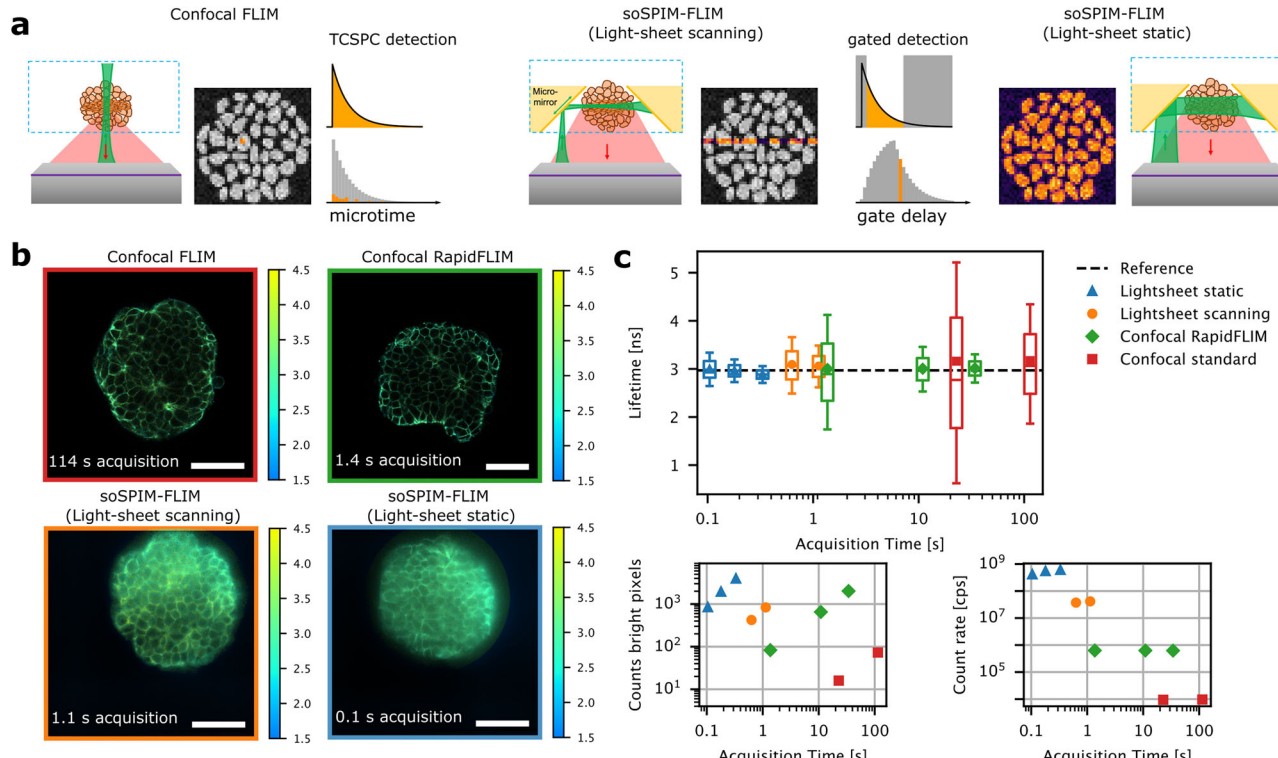

**Fig. 1 | Benchmarking of soSPIM-FLIM in 3D embryonic organoids. a** Scheme of the FLIM modalities: In confocal FLIM (left), photon arrival times are detected in every pixel of the image via TCSPC detection. This generates the lifetime histogram. In soSPIM-FLIM, the sample is illuminated either using a digital scanned (center) or a static light-sheet (right). Photons are recorded through gated detection, in which multiple images are recorded with different gate delays. **b** Exemplary FLIM images of fixed embryonic organoids stained with E-cadherin-AF488, acquired with a conventional or RapidFLIM confocal microscope (top row) or soSPIM-FLIM in scanning or static light-sheet mode.

Measurements were performed at 30–50 μm depth. Acquisition times are indicated. Scale bars are 50 μm. **c** Box plots of pixel lifetimes obtained for the measurements shown in (**b**), pooled from all pixels within the organoid. Symbols denote mean and lines median values. Boxes show the interquartile ranges (IQR) and whiskers 25%–0.5*IQR/75%+0.5*IQR. For confocal data, an increasing number of frames were sampled from the same measurement. The reference is the intensity-weighted average lifetime from sum fits of lower power confocal measurements (Fig. S3). The insets show the counts in a typical bright pixel (99.9% intensity) and the average overall count rate during the image acquisition.

shorter illumination of each pixel row, which leads to local saturation of gates with the highest photon count rate in pixels that are illuminated during the passage of the beam. To account for this, we employed an adapted pile-up correction[25] to the scanning light-sheet data (see "Methods" and SI for details). To evaluate the pile-up effect and the dynamic range of soSPIM-FLIM, we analyzed the lifetime as a function of pixel count rate for different illumination powers (Figs. S1, S6 and S7). Although the adapted correction mitigates the impact of pile-up, static light-sheet still achieves ~10× faster acquisition than scanning light-sheet at comparable photon counts (Fig. 1 and Table S1).

The fast speed of soSPIM-FLIM makes it ideally suited for 3D acquisitions. We performed 3D FLIM measurements in organoids across an 80 μm range, with one image plane every 2 μm, illuminated for 180 ms in static mode and 1.1 s in scanning mode (Fig. 2a). While the average pixel lifetime was constant across depth, we observed a ca. 8% discrepancy between lifetimes determined from static and scanning light-sheet measurements. Compared to the confocal reference, the scanning light-sheet overestimates lifetimes by ~5%. This is explained by residual pile-up in scanning light-sheet in high intensity pixels, since the lifetimes match when only low-intensity pixels are included in the analysis (Fig. S9). Regarding overall image quality, confocal images exhibit superior sectioning and axial resolution, indicated by a higher discrete cosine transform Shannon entropy (DCTS) image quality score[26] (Fig. S8). Axial resolution is sub 1 μm in the confocal microscopes and ca. 2 μm at the waist in the light-sheet microscope (see "Methods"). Nonetheless, soSPIM-FLIM benefitted from less photon loss at deeper z planes compared to confocal FLIM, which showed up to threefold intensity loss towards the center of the organoids at 50 μm depth (Fig. S8). Of the two light-sheet modalities, scanning light-sheet images exhibit a less severe drop of image quality with depth and a higher overall DCTS score (Fig. S8). Generally, soSPIM-FLIM images showed lower optical sectioning and degradation of image quality along the propagation direction of the illumination (Figs. 2 and S8). In light-sheet microscopy, this is commonly circumvented by illuminating the specimen from multiple

sides[27,28]. This can be readily implemented in the JeWell architecture by sequential illumination from two opposing sides of the JeWell mirrors and fusion of the acquired image stacks to a 3D FLIM image stack, with more uniform image quality (Figs. 2b, and S8).

A powerful application of FLIM is to discriminate spectrally overlapping fluorophores by their fluorescence lifetime, providing an additional contrast for multi-target imaging[2,29]. In light-sheet microscopy, lifetime-based multiplexing could allow simultaneous imaging of multiple targets that would otherwise be imaged sequentially and simplify microscope set-ups requiring multiple laser lines and cameras. To test lifetime unmixing, we stained the cell membranes of live embryonic organoids expressing nuclear H2B-green fluorescent protein (GFP) fusion protein with Flipper-TR (Fig. 3a), a live-cell compatible mechanosensitive membrane tension sensor recently applied to in vivo specimens[30] with similar excitation wavelength as GFP but comparably long lifetime[4]. To minimize potential variations of Flipper-TR lifetimes due to heterogeneities in membrane tension, organoids were examined 48 h after aggregation, at which cellular composition is expected to be homogenous[31] and Flipper-TR can be considered as a label with approximately constant lifetime[32], with respect to the expected lifetime difference to GFP[33]. We performed 3D soSPIM-FLIM in scanning mode. Lifetime images obtained with soSPIM showed a clear contrast between nuclei and membrane pixels (Fig. 3b, c and Video S1). To extract the characteristic lifetimes for both species, we selected two populations in the phasor plot[34,35] (Fig. 3a). We then mapped selected points back to the original image, providing masks that showed clear correspondence to either nuclei or membrane pixels. Fitting a double-exponential model to the summed decays provided a short and long lifetime component for each species. We performed this analysis in 3 organoids at 3 different depths (10, 30, 50 μm) and obtained robust lifetime estimates of 2.1 ± 0.2 ns, 5.4 ± 0.9 ns for nuclei (i.e., GFP) and 1.0 ± 0.2 ns, 4.8 ± 0.2 ns for membranes (i.e., Flipper-TR) (Fig. 3b). These values are in good agreement with confocal lifetimes determined from FLIM images acquired for 4 min on the same organoids by manual ROI selection and two-component fitting (Fig. S10),

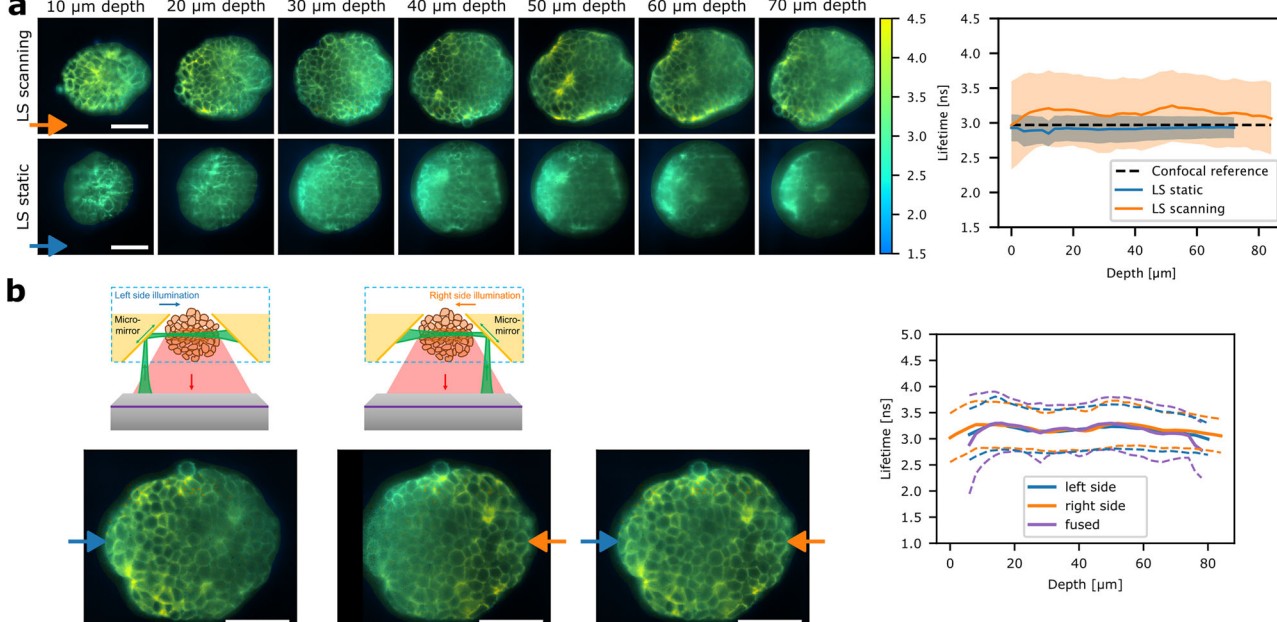

**Fig. 2 | Volumetric (dual-illumination) soSPIM-FLIM in embryonic organoids. a** soSPIM-FLIM images obtained in 3D acquisitions with scanning or static light-sheet with 1.1 s (scanning) or 0.18 s (static) exposure time per plane. The illumination side is indicated by orange/ blue arrows. 3D soSPIM-FLIM stacks were acquired over a range of 80 μm with 2 μm spacing. The graph on the right shows the average lifetime (solid lines) and standard deviation (shaded area) over the whole image as a function of imaging depth. The dashed black line shows the lifetime of the

confocal reference (see Fig. 1c). **b** Dual-illumination soSPIM-FLIM images at 40 μm depth obtained from sequential 3D acquisition from opposing JeWell mirrors at the left or right side of the image, as indicated by blue/ orange arrows and schemes on the top. Image stacks were registered and fused to obtain a final FLIM image with a more uniform image quality (image on the right). Right: Average lifetime and standard deviation as a function of image depth for both illumination sides and the fused image. Scale bars are 50 μm.

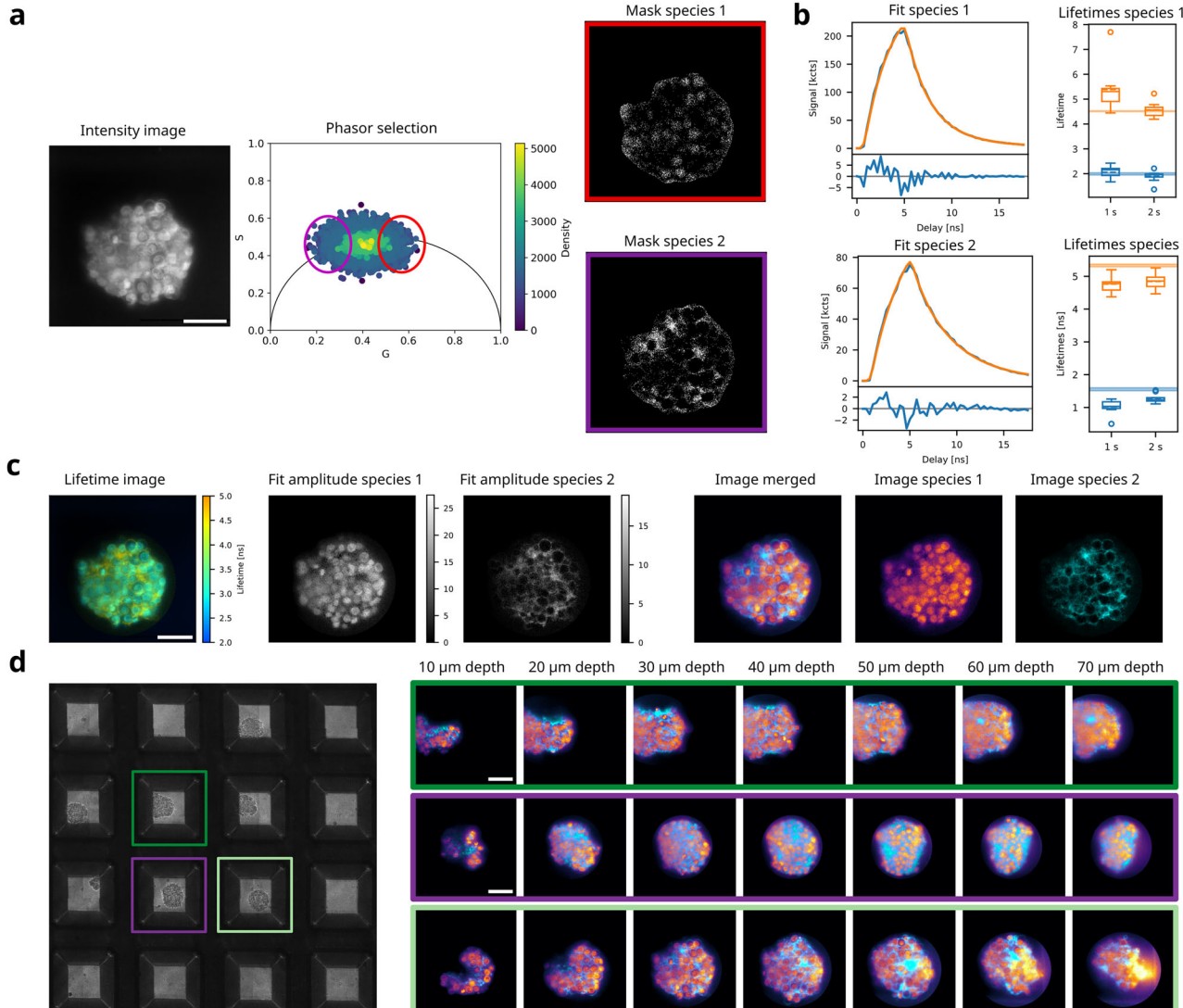

**Fig. 3 | Application of soSPIM-FLIM for lifetime multiplexed light-sheet imaging. a** Pipeline for generation of lifetime patterns for organoids expressing H2B-GFP and additionally stained with Flipper-TR (grayscale image). To identify the two species, soSPIM-FLIM data were transformed into phasor space and points selected using elliptic cursors drawn around the extremities of the point cloud. Backmapping selected points to the image generates two masks. **b** Two-component fitting was performed to extract characteristic lifetimes for the two species. Box plots show lifetimes obtained in three different organoids at three different $z$ planes from measurements with acquisition times of 1 and 2 s in scanning mode. Horizontal lines indicate reference lifetime values extracted from confocal FLIM performed on the same samples (Fig. S10). **c** Lifetime unmixing for H2B-GFP/ Flipper-TR organoids using lifetime patterns described in (**b**). Pixel-wise fitting with the determined lifetime patterns results in species amplitudes, which are used to generate unmixed intensity images (right), showing successful decomposition into nuclei and membrane signal. **d** Composite images obtained by unmixing of slices from 3D soSPIM-FLIM data on the samples described in (**a, b**), imaged across 70 μm with 2 μm spacing and 1 s acquisition time per plane. Shown are three different organoids from a multi-position acquisition on one JeWell chip (brightfield image of wells shown on the left). Scale bars are 50 μm in all images.

validating that the pixels selected in the phasor plot represent pixels with predominant GFP or Flipper-TR signal. In the case of more overlapping structures, reference measurements should be performed in samples stained with only one fluorophore species to determine individual lifetime patterns or confirm the phasor positions. The slightly lower lifetimes measured for Flipper-TR with soSPIM-FLIM are likely a consequence of the lower excitation wavelength (i.e., 488 nm) compared to confocal FLIM (i.e., 510 nm), as recently shown[36]. Finally, we fitted a superposition model of both lifetime patterns to each pixel, providing amplitudes that we then used to calculate unmixed intensity images (Fig. 3c). With this pipeline, we could well separate nuclei and membranes, i.e., GFP and Flipper-TR signal, which is only separated by 1.2 ns in average lifetime (see SI). Lifetime unmixing worked reliably in different $z$ planes across 70 μm depth in multiple organoids (Fig. 3d and Video S1). We hereby benefited from the multi-position imaging capabilities of soSPIM[19], enabling us to perform soSPIM-FLIM on

multiple organoids located in the same JeWells chip (Fig. 3d) without the need for time-consuming re-alignment or repositioning of the beam for each organoid[19].

The speed and gentleness of soSPIM-FLIM make it an ideal tool for volumetric time-lapse FLIM on sensitive, dynamic specimens, which is strongly limited with confocal FLIM due to requirements for either high excitation powers or long acquisition times to achieve sufficient photon counts. As a proof-of-concept, we performed 3D time-lapse soSPIM-FLIM on live embryonic organoids stained with Flipper-TR. Due to the limited photon budget in confocal FLIM, its lifetime is usually determined globally across many cells acquired in one plane for several minutes[32]. With soSPIM-FLIM, we could robustly quantify Flipper-TR lifetimes across $z$-stacks with only 1–3 s acquisition time per 2D plane (Fig. 4a, b). In agreement with previous studies[32], we obtained two characteristic lifetimes of 1.0±0.1 ns and 5.0±0.1 ns, averaged over 3 organoids and 18 $z$ planes each (Fig. 4b and

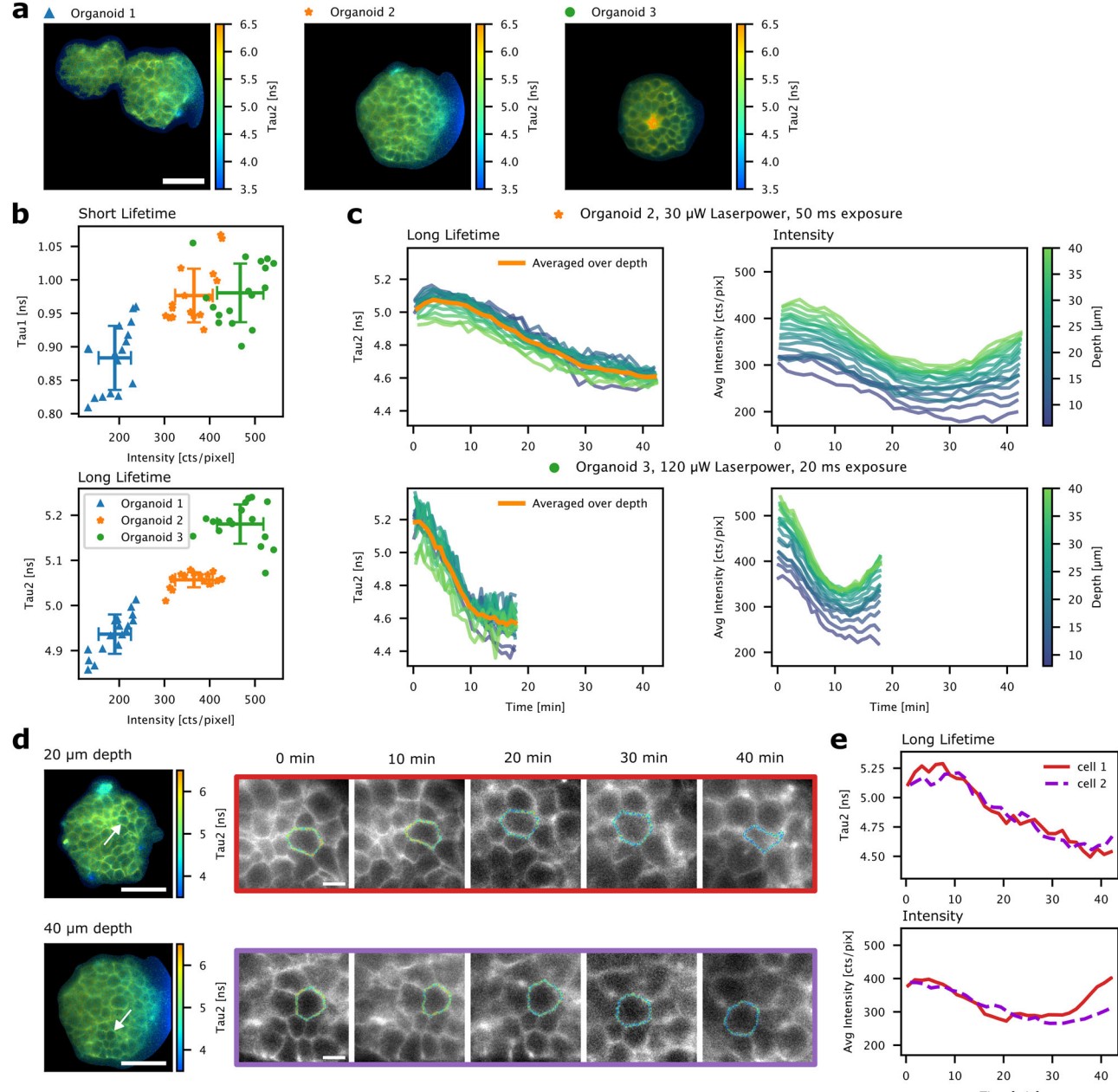

**Fig. 4 | Application of soSPIM-FLIM for live volumetric time-lapse membrane tension imaging in organoids. a** FLIM images of Flipper-TR stained live organoids at a depth of 30 μm. The color scale shows the long lifetime from a two-component fit with the short lifetime fixed (see "Methods"). 3D time-lapse FLIM imaging of organoids 2 and 3 (**a**) was performed with different laser power and exposure, with one $z$-stack (30 planes) acquired every 1.5 min for organoid 2 and every 45 s for organoid 3. **b** Fit results from two component sum fits at the start of the time-lapse acquisition. Sum decays of segmented membrane pixels (Fig. S11) in each $z$-plane at 10–40 μm depth were fitted with two free lifetimes. Each point is the result from one $z$-plane, with averages and standard deviations for each organoid shown as error bars. **c** Long Flipper-TR lifetime and average membrane pixel intensity across multiple $z$-planes over the time-lapse acquisition of organoids 2 and 3. In every $z$-plane, sum decays of segmented membrane pixels were fitted with a two-component fit with the short lifetime fixed. **d** Tension imaging of individual cells in organoid 2. Lifetime images on the left show the long lifetime at 20 μm and 40 μm depth at 0 min (see "Methods"). Cropped images show two individual cells (arrows in the images on the left) that were segmented and tracked over time. Intensity is displayed in grey and long lifetime in color code along the cell contour. **e** Long Flipper-TR lifetime and average intensity from a sum fit over segmented cell pixels. The analysis was restricted to a single 2D plane per cell. Scale bars are 50 μm in all images showing entire organoids and 10 μm in the cropped images in (**d**).

Table S2). While the mean short lifetime only varies by about 100 ps between organoids, we observed larger variance in the long lifetime, up to 250 ps. Potential reasons could be residual pile-up or illumination-induced lifetime changes, although no clear count rate dependence within organoids or power dependence between organoids are observed (Fig. S11 and Table S2). A more likely explanation could be biological variance, for instance due to the visible lumen in organoid 3. We repeatedly acquired $z$-stacks of 2 organoids over a course of 45 and 17 min, respectively, to analyze

changes in the long lifetime ($\tau_2$) over time in each $z$-plane. In both organoids, we observed a decrease in $\tau_2$ over time (Fig. 4c), which is in line with a decrease in membrane tension[32]. Additionally, we observed an initial decrease followed by a recovery in average membrane pixel intensity. Interestingly, these dynamics are accelerated and increased in amplitude with higher illumination power (Fig. 4c). This power dependence indicates that the observed lifetime change might be a consequence of illumination-induced photosensitization of Flipper-TR, i.e., formation of reactive oxygen

**Table 1 | Comparison of light-sheet FLIM parameters and performance presented in this work to previous light-sheet FLIM implementations**

| Work | Detection objective | Detector | Illumination configuration | FOV [pix] | Fastest frame time [s] | Pixel count rate [Hz] | Overall count rate [MHz] |
|---|---|---|---|---|---|---|---|
| Hirvonen et al.[10] | 20× 0.5 | Photek MCP | Static | – | 5 | – | 0.032 |
| Mitchell et al.[9] | 20× 0.5 | PCO.FLIM | Scanned | 1008 × 1008 | 4 | – | – |
| Samimi et al.[13] | 10× 0.3 | Horiba Flimera | Static | 192 × 128 | 10 | 300 | 7.5 |
| Nedbal et al.[11] | 60× 1.49 | Horiba Flimera | Static | 192 × 128 | 1 | 284 | – |
| Li et al.[8] | 10× 0.3 | Intensified CMOS | Scanned | 2048 × 2048 | 2 | – | – |
| This work | 40× 1.2 | SPAD512s | Static | 512 × 512 | 0.1 | 8100 | 414 |
| | 40× 1.2 | SPAD512s | Scanned | 512 × 512 | 0.6 | 700 | 37.2 |

species and lipid oxidation which change the properties of the plasma membrane upon continuous illumination, as recently reported[37], or phototoxicity and the associated cellular response. We furthermore tracked the Flipper-TR lifetime $\tau_2$ in individual, fastly rearranging cells from, on average, 90,000 photons detected per cell (ca. 15,000 per junction) (Figs. 4d and S12, and Video S2). The time dependence of $\tau_2$ and average intensity observed in single cells reproduce the time dependence found for the sum fits averaged over many cells. This shows that lifetime changes of a few 100 ps can be robustly tracked in single cells over 45 min, even though the applicability of this for tension sensing is limited by the illumination-induced lifetime change.

## Discussion

In this work, we have realized fast, optically sectioned wide-field FLIM by integration of pulsed excitation and time-resolved detection into soSPIM. The soSPIM-FLIM technique achieves 10–100× speed advantage compared to state-of-the-art confocal FLIM, with excellent quantitative agreement of measured lifetimes. We have demonstrated volumetric lifetime multiplexed light-sheet imaging and volumetric tension imaging on individual cells, opening up new possibilities for functional imaging on living samples such as embryos and organoids.

We have benchmarked soSPIM-FLIM in both static and digital scanned light-sheet modalities, making the results transferable to a broad range of light-sheet systems. Our benchmark reveals a trade-off between FLIM performance and image quality in challenging 3D samples between both modalities: Speed and dynamic range of FLIM with scanned illumination are limited by detector pile-up at currently available scan rates, but nevertheless provided robust lifetime multiplexed imaging in 3D and superior image quality in deeper tissue layers (Fig. S8). Static light-sheet illumination, however, has clear advantages with respect to speed and pile-up over scanned illumination when combined with the time-gated detector architecture employed here, at the expense of reduced image quality in scattering samples, as previously discussed[38].

While such considerations regarding the interplay between image quality, illumination and sensor architecture are required, the simplicity of adding pulsed excitation and a time-resolved camera to a light-sheet microscope will enable FLIM in new contexts in the future. The FLIM performance of the here presented system represents a substantial progress for fast, volumetric FLIM compared to previous light-sheet implementations, benefitting from much faster speed, higher detection NA (except the previous work by Nedbal et al.[11]), and so far unachieved pixel and overall count rates (Table 1).

In conjunction with our recently presented high-throughput 3D imaging pipeline[19], soSPIM-FLIM can be automated to achieve high-content 3D FLIM imaging, at a throughput of about 50 organoids per hour. While this work has focused on the soSPIM geometry, i.e., a particular light-sheet architecture employing 45° reflective mirrors and a single objective for illumination and detection, our implementation is fully compatible with other, single- or multi-objective light-sheet configurations[27,39–42], provided sufficient light collection efficiency can be achieved. In this context, FLIM implementation on state-of-the-art OPM systems[41,42] achieving high effective detection NA through remote focusing and volumetric acquisition by fast galvo scanning will be particularly promising. Nevertheless, high-speed volumetric imaging will be restricted in FLIM mode by the minimum acquisition time required to detect meaningful lifetimes with gated read-out. To explore this limit quantitatively, we have sub-sampled the fastest acquisitions presented in this work (static light-sheet FLIM with 100 ms acquisition, i.e., 25 gates with 3 ms exposure per gate). Robust lifetimes can be achieved at ca. 65 ms acquisition time (Fig. S13), with compromises or restriction to ratiometric/ relative readouts even below, approaching 2D acquisition times only reached by compressed sensing methods[16] or Pockels cell gating [14,15]. Thus, combining high-speed OPM imaging (typically operating with static sheet illumination) with time-gated detection could facilitate FLIM imaging of dynamic (sub-)cellular processes at volumetric rates above 1 volume-per-second.

With the adapted pile-up correction demonstrated in this work, FLIM with gated detection becomes compatible with scanned illumination, enabling additional light-sheet illumination schemes such as two-photon excitation[43] or scanned Bessel beams [44]. The compatibility with multi-view light-sheet systems enables adapting light-sheet FLIM to bigger specimens, a potential illustrated by our dual-illumination soSPIM-FLIM.

Notably, light-sheet FLIM operates at orders of magnitude lower peak illumination powers than confocal FLIM. We took advantage of this with proof-of-concept 3D time-lapse membrane tension imaging on single cells in living embryonic organoids. Interestingly, we observed a lifetime decrease over time, which we attribute to changes of the plasma membrane environment, induced by repetitive illumination of the Flipper probe[37]. In the context of this and other live imaging applications of light-sheet FLIM, future work will be needed to evaluate the viability of specimen exposed to prolonged pulsed illumination for different samples, fluorophores, and experimental conditions.

Overall, light-sheet FLIM not only massively accelerates FLIM imaging, but also provides new quantitative readouts for light-sheet microscopy beyond fluorescence intensities. We anticipate applications in dynamic (multi-)cellular specimen currently restricted to 2D FLIM imaging with laser scanning microscopes to generate 3D maps of, e.g., molecular forces and protein-protein interactions via FLIM-FRET[45,46], metabolic gradients via lifetime-based label-free imaging[47], or concentrations of, for instance, oxygen or calcium via FLIM-based sensors[33]. Applied to living organisms and the growing number of organoid models, this approach could thus become a crucial tool for advancing our understanding of physiology in both health and disease.

## Materials And methods
### Microscope setups: soSPIM-FLIM
Light-sheet imaging was performed on a custom-made soSPIM microscope built around a Zeiss Axiovert 200 microscope body (Carl Zeiss) following the previously published design[17]. It consists of a custom laser scanning unit, dedicated microfabricated devices[19] containing an array of truncated pyramidal-shaped microwells flanked with 45° mirrors (JeWells), and a

soSPIM acquisition module operating in MetaMorph (Molecular Devices). The microscope is equipped with an LED lamp (CooLED, Andover) for brightfield imaging, continuous-wave excitation lasers at 488, 514 and 561 nm and a sCMOS camera (Hamamatsu ORCA Flash 4.0 V3). The sample is illuminated and fluorescence is detected through a 40× 1.2 NA Plan Apochromat water immersion objective (Carl Zeiss). The light-sheet was generated by either scanning the laser beam laterally along the JeWell mirrors or inserting a cylindrical lens ($f = 150$ mm, Thorlabs) into the optical path and aligning it according to the JeWell mirrors' axis[17,18]. 3D imaging is achieved by moving the beam perpendicular to the mirror axis through a pair of galvanometric mirrors (Pangolin SCANMAX 506 actuators) while translating the objective axially using a piezo-driven focus controller (Physik Instrumente). The resulting displacement of the light-sheet waist is compensated by shifting it using an electrical tunable lens (Custom EL-30–10, Optotune). The light-sheet position (axial position and waist) is synchronized with the focal plane of the objective thanks to the soSPIM acquisition module, which ensures optimal 3D sectioning. The dimensions of the light-sheet were controlled by adjusting the laser beam diameter using a tunable diaphragm placed after the fiber output of the coupled excitation lasers. The resulting dimensions for scanning and static light-sheet were ca. 2.2 μm/1.9 μm thickness (full width half maximum) at the waist and ca. 40 μm/30 μm field of view (twice the Rayleigh length). The width of the static light-sheet was ca 110 μm.

For FLIM detection, a 512 × 512 pixel SPAD array detector (SPAD512[2], Pi Imaging Technology) was mounted on the second port of the microscope. The image plane of the detector was aligned with the sCMOS camera and centered on the same field of view. The effective field-of-view of the SPAD array is 210 × 210 μm$^2$ with a pixel size of 410 nm. For FLIM excitation, a high power pulsed 488 nm laser prototype[21] (PicoQuant GmbH) was fiber coupled into the optical path. Time gating of the SPAD array detector was triggered by the laser via a NIM to TTL converter (TMF400, PicoQuant GmbH). For 3D FLIM imaging, image acquisition on the SPAD array was synchronized with soSPIM illumination using a TTL trigger signal. To achieve dual-illumination imaging from two sides, two opposing mirrors of a JeWell were illuminated sequentially. The sample was shifted in between to fit it into the field of view of the SPAD array detector. For each side, the waist of the light-sheet was positioned at a distance of about one-quarter of the specimen's thickness from the reflecting mirror. To image multiple organoids in different positions, the previously described tools for JeWell chip preview and positioning of individual JeWells were employed using bright field imaging[19]. For live imaging, the JeWell chip was placed into an incubation chamber (Okolab), which was mounted on top of the microscope stage, heated up to 37 °C and provided with 5% $CO_2$ and 85% humidity.

## Microscope setups: confocal FLIM

FLIM measurements were performed on a Zeiss LSM880 system (Carl Zeiss) equipped with a time-resolved LSM upgrade (PicoQuant GmbH) using a Plan-Apochromat 40× 1.2 NA water immersion objective. Images of 512 × 512 pixels per frame were acquired after excitation with a 510 nm pulsed laser diode operating at 20 MHz repetition rate. Fluorescence was split using a 560 nm dichroic mirror and detected by two PMA Hybrid-40 detectors (PicoQuant) after passing 550/49 nm and 600/50 nm bandpass filters, respectively. The pinhole was set to 1 airy unit. In each measurement, images were accumulated over a time period of 2 min, with a pixel dwell time of 1.23 μs and frame time of 0.6 s.

## Microscope setups: confocal Rapid FLIM

Rapid FLIM Measurements were performed on a MicroTime 200 confocal microscope (PicoQuant GmbH) equipped with a LDH-D-C-485 picosecond excitation laser, a FLIMBee beam scanner, a PMA Hybrid-40 detector, and a MultiHarp 150 TCSPC unit[48] (all PicoQuant GmbH). RapidFLIM images of fixed organoids stained with E-cadherin-AF488 were acquired with a 60× 1.2 NA water immersion objective (UPLSAPO60XW, Olympus), with 485 nm pulsed excitation at a 40 MHz laser repetition rate with 512 × 512 pixels, 500 nm pixel size, and a dwell time of 1.3 μs.

## Data acquisition

Light-sheet FLIM data acquisition was performed with the Pi Imaging software in streaming mode, controlled by self-written Python scripts. Acquisition parameters are listed in Table S2. The light-sheet microscope was controlled via MetaMorph (Molecular Devices). Confocal microscopes were controlled with ZEN (Carl Zeiss) and SymPhoTime64 (PicoQuant) in the case of the LSM880 and with SymPhoTime64 in the case of the MicroTime 200.

## Data analysis: instrument response function

The combined instrument response function (IRF) of the SPAD array and the laser was characterized by detecting reflected laser light from a mirror placed on the sample holder. The measured IRF is parametrized with an approximated function, given by integration of a Gaussian laser IRF with an infinitely sharp gate edge:

$$IRF(t) = \sqrt{\frac{\pi}{2}}\sigma\left[erf\left(\frac{t-t_0+w}{\sqrt{2}\sigma}\right) - erf\left(\frac{t-t_0}{\sqrt{2}\sigma}\right)\right]$$

with the gate width $w$, the width of the laser IRF $\sigma$, and the gate delay offset $t_0$ as free parameters. With this IRF model the sigma parameter compensates for the finite edge steepness of the SPAD array gating function, and is a measure for the combined edge steepness given by the laser pulse width and the detector gating function. By parametrizing the IRF, relative delays between the IRF and light-sheet measurements can be compensated and the high demands for experimental IRF data quality are relaxed (Fig. S14).

## Data analysis: background and pile-up correction

To limit the impact of dark counts, background measurements (Fig. S14) were taken daily with the same settings as the fluorescence measurements and subtracted from measured data. To account for detector saturation, we employed an adapted pile-up correction[49]. The corrected photon number in 8-bit images is calculated from the measured photon number as:

$$N_{corr} = -\ln\left(1 - N*F/255\right)*255/F$$

with $F = 1$ for static light-sheet measurements and $F = 3.5$ for scanning light-sheet measurements. Pile-up correction was applied after background correction. A detailed discussion of pile-up correction can be found in the SI.

## Data analysis: fitting

Time-gated FLIM signals are fitted with an approximate model function. This is based on an analytic approximation function for the convolution of a Gaussian pulse with an exponential decay, according to[50]:

$$f(t) = \frac{1}{2}exp\left(\frac{\sigma^2}{2\tau^2}\right)\left(1 - erf\left(\frac{\sigma}{\sqrt{2}\tau} - \frac{t}{\sqrt{2}\sigma}\right)\right)exp\left(-\frac{t}{\tau}\right).$$

where $\sigma$ is the width of the Gaussian pulse and $\tau$ is the monoexponential decay time. The model function for decays obtained by time-gated detection is then obtained by integrating the decay function with infinitely sharp gate edges, analogous to the model function for the IRF:

$$\begin{aligned} f_{Gated}(t) = {}& A\frac{\tau}{2}exp\left(\frac{\sigma^2}{2\tau^2}\right)\left[\left(erf\left(\frac{\sigma}{\sqrt{2}\tau} - \frac{t-t_0+w}{\sqrt{2}\sigma}\right) - 1\right)exp\left(-\frac{t-t_0+w}{\tau}\right)\right. \\ & \left. + exp\left(-\frac{\sigma^2}{2\tau^2}\right)erf\left(\frac{t-t_0+w}{\sqrt{2}\sigma}\right)\right] \\ & - A\frac{\tau}{2}exp\left(\frac{\sigma^2}{2\tau^2}\right)\left[\left(erf\left(\frac{\sigma}{\sqrt{2}\tau} - \frac{t-t_0}{\sqrt{2}\sigma}\right) - 1\right)exp\left(-\frac{t-t_0}{\tau}\right)\right. \\ & \left. + exp\left(-\frac{\sigma^2}{2\tau^2}\right)erf\left(\frac{t-t_0}{\sqrt{2}\sigma}\right)\right] \end{aligned}$$

where $A$ is the decay amplitude, $t_0$ can express a time offset between the gate delays and the laser pulse and $w$ is the width of the approximated gate. The use of this analytic model function without explicit convolutions not only

speeds up pixel-wise fitting, but also simplifies the use of the parametrized IRF and thus the compensation of IRF variations across the detector area (see SI). The IRF parameters $\sigma$ and $w$ were held fixed at the extracted pixel values from the IRF measurement for all fits. Maximum likelihood estimation of model parameters from experimental decay signals was performed with the L-BFGS-B implementation within the scipy optimize package.

### Data analysis: dual-illumination registration and fusion

Image stacks acquired from two opposite illumination sides were first registered in 3D using the Blockmatching algorithm[51] based on a 3D translation. In the registered image stacks, aggregates were segmented in a maximum intensity projection after Gaussian blurring with a sigma of 10 pixels, mean thresholding and dilation of the obtained mask with disk of radius 10. A bounding box surrounding the union of the two segmentation masks was then calculated and fusion weights $w_{1,2}$ defined using a sigmoid function,

$$w_{1,2}(x) = \frac{1}{e^{\pm \frac{x - l/2}{l/8}} + 1}$$

centered around the mid-line of the bounding box, where $l$ is the width of the bounding box, $x$ the spatial coordinate along the light propagation axis. Image and lifetime stacks were then fused using a weighted sum[52,53],

$$I(x,y,z) = \frac{w_1(x)I_1(x,y,z) + w_2(x)I_2(x,y,z)}{w_1(x) + w_2(x)}$$

$$\tau(x,y,z) = \frac{w_1(x)\tau_1(x,y,z) + w_2(x)\tau_2(x,y,z)}{w_1(x) + w_2(x)}$$

where $I_{1,2}$ and $\tau_{1,2}$ are the intensity and lifetime stacks acquired from each illumination side and $(x,y,z)$ denote the spatial coordinate of each image voxel.

### Data analysis: image quality estimation

To quantify the intensity loss towards the center of the organoid in confocal imaging (Fig. S8), segmentation masks were generated with Gaussian blurring and otsu thresholding. Intensity and shortest distance to the mask edge were computed for every pixel in the mask. Pixels were pooled every 2 μm of distance and the intensity of the 99th percentile was taken as a measure for a typical bright pixel within that distance.

Image quality in 2D confocal/ light-sheet cropped images and 3D static/ scanning light-sheet stacks (Fig. S8) was quantified using the normalized DCTS entropy[26]. Cropped images were obtained by segmenting organoids using Gaussian blurring and Otsu thresholding and taking the central $128 \times 128$ pixels of the segmentation mask. To quantify image quality along the propagation direction of the light-sheet for dual-illumination data, DCTS was quantified in four rectangular stripes as a function of imaging depth.

### Data analysis: phasor analysis

Phasor plots were generated using a discretized phasor transformation[35]. From the detected lifetime decay $N_{m,ij}$ in each pixel ($N_{m,ij}$: photon count in gate delay $m$ in pixel $ij$), the phasor coordinates were calculated as follows:

$$G_{ij} = \frac{\sum_{m=0}^{k-1} N_{m,ij} \cos\left(\omega\left(m + \frac{1}{2}\right)\Delta T\right)}{\sum_{m=0}^{k-1} N_{m,ij}}$$

$$S_{ij} = \frac{\sum_{m=0}^{k-1} N_{m,ij} \sin\left(\omega\left(m + \frac{1}{2}\right)\Delta T\right)}{\sum_{m=0}^{k-1} N_{m,ij}}$$

where $\omega = 2\pi/T$, with the total detection time $T$, the gate width $\triangle t$ and $k$ the number of gate delays. To account for the IRF, the total phasor and the phasor of the IRF were calculated in complex space and the IRF phasor divided from the total phasor in each pixel. Afterwards, the real and imaginary parts $G_{ij}$ and $S_{ij}$ were determined.

### Data analysis: pattern matching

To decompose FLIM images on organoids containing two fluorophore species, i.e., GFP and Flipper-TR, phasor plots of FLIM images were generated in three different $z$ planes (i.e., at 10, 30 and 50 μm depth) for three different organoids. In each phasor plot, two elliptic cursors were drawn around the extremities of the point cloud and selected points mapped back to the image to define pixel masks for individual fluorophore populations. Lifetime decays were then summed across all corresponding pixels and a double-exponential decay model fitted to the overall decay for each species. Averaging over all organoids and $z$ depths, this procedure yielded two characteristic lifetimes and respective amplitudes for each species, defining their lifetime patterns. Next, a superposition model, defined as the sum of the two lifetime patterns (i.e., two double exponential decay functions, each with fixed decay times and amplitudes) was fitted pixel-wise to the lifetime decay in each pixel of the 3D FLIM stack with species amplitudes as free fit parameters. Final intensity images for each species were generated by weighing the total pixel intensity in the original image with the determined species amplitudes. Before intensity weighting, the amplitudes were smoothed with a Gaussian kernel (sigma=1 pixel) to reduce pixel-to-pixel variations.

### Data analysis: flipper-TR time-lapse analysis and cell segmentation

For Flipper-TR measurements, FLIM analysis was performed in two iterative rounds of fitting with a two-component decay model. First, each 2D plane was intensity thresholded, using pixels between the 50th and 95th intensity percentile to select in focus areas and exclude very bright areas to reduce the impact of pile-up. Detected lifetime decays of remaining pixels were summed and a double-exponential decay model fitted to the overall decay, with both lifetime components and the amplitude ratio as free fit parameters. Then, the average short lifetime (1.0 ns) and amplitude ratio (0.45) from all $z$ planes across 3 organoids was determined (Fig. 4). For all subsequent analysis of changes in the long lifetime ($\tau_2$), the short lifetime and amplitude ratio were fixed to these values. For individual cell lifetime analysis of 3D time-lapse Flipper-TR data, cells of interest were selected and, for each time point and cell, the image containing the cell at its central $z$-position extracted from the $z$-stack. Cell membranes were segmented in these images in 2D, using the Tissue Analyzer plugin[54] in Fiji[55]. Label images containing a separate label for each segmented junction were manually curated in Napari[56] and junctions belonging to the same cell pooled to obtain segmentation masks for each time point. Subsequently, cells masks were manually annotated in time. Finally, masks were dilated with a disk kernel of radius of one pixel to account for the width of the cell outlines in the images.

### Data analysis: confocal FLIM

All confocal FLIM data were analyzed using the SymPhoTime64 software (PicoQuant GmbH, Berlin, Germany). For pixelwise single exponent estimation (Ecad-AF488) the fast lifetime algorithm was used, corresponding to center of mass analysis. For reference lifetimes of single fluorophore data, bi-exponential re-convolution fitting of sum signals at moderate intensity was performed (Fig. S3). For reference lifetimes of dual-labeled samples, decay patterns were generated from manually selected ROIs (Fig. S10) of nuclei and membranes. Sum decays of the ROIs were fitted with a bi-exponential re-convolution fit to obtain the reference lifetimes given in Fig. 3.

### Sample preparation: solution measurements

Alexa Fluor 488 (AF-488) NHS Ester (ThermoFisher Scientific) was dissolved in PBS at 200 nM and pipetted onto the soSPIM chip for imaging.

### Sample preparation: gastruloid culture, staining and mounting

A complete description of the culture conditions and the protocol for making gastruloids is presented in Baillie-Johnson et al.[57]. In the experiments presented here, gastruloids were generated from E14Tg2a.4 or H2B-GFP mouse embryonic stem cells (mESC) (MMRRC, University of California Davis, US) using an initial cell number of 100, and cultured according to the modified protocol described in Hashmi et al.[31]. After 2 days of culture, gastruloids were incubated for 1 h in culture medium supplemented with 1 μM Flipper-TR and manually transferred to JeWell chips with 120 or 170 μm well size (top opening). E14 gastruloids were fixed using 4% paraformaldehyde and immunostained for E-cadherin using rat primary antibody and anti-rat-AF-488 secondary antibody, following published protocols[31]. JeWell culture chips were rinsed with 96% ethanol, passivated with 0.2% lipidure solution and rinsed in PBS after evaporation. To remove air bubbles, chips were evacuated in a vacuum chamber for 1 h. Afterwards, chips were sterilized under UV illumination for 30 min and medium was exchanged to cell culture medium using multiple washes. Live or fixed gastruloids were mounted manually by pipetting under the occular (source) just before imaging. For RapidFLIM measurements, fixed gastruloids were placed in a 35 mm glass-bottom dish (#1.5, MatTek Corp., Ashland, MA).

### Reporting summary

Further information on research design is available in the Nature Portfolio Reporting Summary linked to this article.

### Data availability

Source data underlying graphs can be obtained from Supplementary Data 1. Example raw FLIM data can be found at https://github.com/ValDunsing/soSPIM-FLIM. All other data are available from the corresponding authors on reasonable request.

### Code availability

Analysis code is available on GitHub at https://github.com/ValDunsing/soSPIM-FLIM.

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

## Acknowledgements

We thank Agathe Rostan (IBDM Marseille) for help with sample preparation, Frank Schnorrer (IBDM Marseille) for access to the confocal FLIM system. We thank Daniel Sapede and Brice Detailleur from the IBDM imaging facility for technical assistance. V.D.-E. acknowledges support by an HFSP long-term postdoctoral fellowship (HFSP LT0058/2022-L). This work is supported by the French National Research Agency ("Adgastrulo" project ANR-19-CE13-0022) and the Fondation de la Recherche Médicale (to P.-F.L. EQU202003010407). We also acknowledge the France-Bioimaging Infrastructure (ANR-10-INBS-04). J.H. thanks the Federal Ministry of Education and Research (BMBF) for their financial support of this work in project LIVE2QMIC/13N15953 as part of the program "Enabling Technologies für die Quantentechnologien". J.-B.S. and R.G. acknowledge the ANR-22-CE42-0017 DEEPHEPATOSCREEN and the University of Bordeaux's IdEx "Investments for the Future" program (GPR BRAIN_2030).

## Author contributions

V.D.-E. and J.H. conceived the project idea, coordinated the project, performed all experiments, wrote the analysis code, analyzed and visualized all data, co-wrote the manuscript. V.D.-E., J.H., and C.C. aligned the optical setup. V.D.-E., J.H., and P.-F.L. discussed the results. T.S. and C.N. designed and built the 488 nm laser prototype. L.G. provided software. R.G. provided assistance with the optical setup. G.G. conceived and fabricated the JeWell chips. M.T. assisted with data analysis. R.G., G.G., J.-B.S., and V.V. developed the soSPIM technology. I.M.A. developed the SPAD array detector. F.K., R.E., and P.-F.L. provided funding and equipment. All authors co-edited the manuscript.

## Competing interests

The following authors declare competing interests: J.H., T.S., M.T., F.K., and C.N. are employed at PicoQuant GmbH, R.E. is the CEO and co-founder of PicoQuant GmbH, I.-M.A. is the CEO and co-founder of Pi Imaging Technology. The other authors declare no competing interests.
