## [Transparent Peer Review file · Communications Biology]

Fast volumetric fluorescence lifetime imaging of multicellular systems using single-objective light-sheet microscopy

Corresponding Author: Dr Valentin Dunsing-Eichenauer

Version 0:

Reviewer comments:

Reviewer #1

(Remarks to the Author)

In the manuscript Fast volumetric fluorescence lifetime imaging of multicellular systems using single-objective light-sheet microscopy Dunsing-Eichenauer & Hummert et al. present an approach to light sheet Fluorescence Lifetime Imaging Microscopy utilising a Single Objective Light Sheet (or single objective Selective Plane Illumination Microscopy – soSPIM) configuration. I would firstly like to commend the authors on a well-written and clearly presented scientific work that was enjoyable to read. The manuscript is well-written and easy to understand, and the authors elegantly point out both the strengths and potential limitations of their technology. The authors also present a detailed and quantitative comparison of the performance of their soSPIM approach to specific confocal FLIM versions in terms of imaging speed and signal strength. This specific comparison would benefit from adding a comparison of the axial resolution between the light sheet and confocal approaches.

The optical setup that enables light sheet imaging using a single objective is based on a previous publication and cited as such. To enable FLIM imaging, a pulsed 488 nm laser and a 512x512 SPAD array detector are incorporated into the setup. The single objective light sheet configuration allows optically sectioned FLIM images to be acquired at speeds far surpassing that of confocal-based systems. Numerous works have previously explored ways of combining light sheet with FLIM using both very different but also very similar types of detection devices, as stated and cited by the authors. By enabling the creation of the light sheet and detection of the fluorescence through the same objective, the authors remove the need for additional bulky optical elements around the sample and allow for a high-NA detection objective to be used, but instead introduce the requirement of a highly specialised sample mounting chip to reflect the excitation light and create the light sheet. It is also noted that the previous work cited in ref 13 presents an alternative way to use a high-NA detection objective (1.49 NA) in light sheet FLIM. A brief mention of other approaches to single objective light sheet systems such as SCAPE/OPM and their compatibility with FLIM would also be welcome.

Overall, the manuscript is clear and presents an approach to light sheet FLIM imaging which has not been previously demonstrated, and the alternative approach to a light sheet FLIM system not requiring multiple objectives is attractive. For certain applications that are compatible with the sample chip, the technology may provide a new and impactful readout, but the wide applicability and adoptability of the method will be restricted by the need for highly specialised sample mounting chips, which make up a critical component of the system.

Below are more specific comments on certain points in the manuscript that I hope can help strengthen the work and clarify some points.

In Fig 1, the highest 0.1% of the pixels are excluded from the quantification. A clearer explanation for this would be good.

The sentence starting with "In contrast" on line 148 does not clearly describe what I believe it is intended to say. I would suggest (if I've understood it correctly) re-writing the sentence to more clearly explain that it is the degradation of image quality along the light sheet propagation direction that is discussed.

In Fig 2, the above-mentioned image degradation is supposedly shown, but the figure (a) does not indicate where the light sheet illumination is coming from.

Relating to Fig 2, but also to the manuscript as a whole, I would invite a more in-depth discussion on the potential benefits of the scanned light sheet. As it stands, very few benefits are presented, and it is not entirely clear why it has been included as an alternative method at all.

In the example application, measurements are taken on an organoid expressing H2B-GFP and stained with Flipper-TR. The reference measurements for the individual lifetimes are also taken in an organoid containing both labels. Why not take the reference measurements for the different labels on organoids containing only one or the other label?

In addition to the above, the Flipper-TR is a mechanosensitive label and, at first thought, does not seem like the most suitable label to use in the multiplexing experiment due to the potentially varying lifetime. A more comprehensive discussion as to why this was chosen would help.

In Fig 4, there seems to be quite a large variation in the initial τ_2 lifetimes measured over the three different spheroids. This difference is not clearly explained in the text or figure legend. Could you please comment on this? Is this due to pile-up effects?

In methods, the light sheet waist is stated to be 1.5/1.9 μm with a Rayleigh length of 50/80 μm . These numbers would require some clarification as to their exact meaning, the waist seems thinner than expected from my calculations.

In the legend of figure S2 an intensity loss of 25-30% is mentioned. However, it is very hard to see or understand what these percentages refer to or how they are calculated.

Reviewer #2

(Remarks to the Author)

In their manuscript, Dunsing-Eichenauer et al. used a SPAD array detector to implement fluorescence lifetime imaging with 10-100 times faster acquisition times compared to confocal FLIM. The high imaging speed is a result of the parallel probing of image pixel as compared to the sequential, pixel-by-pixel acquisition with a laser scanning microscope. To demonstrate the capability of their setup, the authors imaged organoids expressing H2B-GFP and stained with Flipper-TR. Using phasor FLIM analysis, the two dye species could be separated from each other. Further, SPIM-FLIM was used for volumetric time-lapse imaging of membrane tension in organoids.

Overall, the data presented seems solid and scientifically sound. High speed FLIM data acquisition with optical sectioning is highly desired for biological studies and biomedical imaging. The manuscript is well written and the experiments are described with sufficient detail. In my opinion, the manuscript is well deserving of publication in Communications Biology after addressing the following comments and questions:

-Figure 2a/Lines 143-144:

There is a 10% systematic deviation between the average pixel lifetime determined from static light-sheet measurements compared to scanning light-sheet measurements. The authors claim that this difference may be a result of the residual pile-up in scanning light-sheet in high intensity pixels. To test this hypothesis, I suggest to perform an analysis with those high intensity pixels excluded from the data.

-Figure 3a:

Two cursors were used to select lifetime pixel on the phasor plot representative of H2B-GFP and Flipper-TR. While the selection makes sense based on the resulting masked images, it would be better to select the position of those cursors based on the expected/measured positions of the lifetimes of H2B-GFP and Flipper-TR on the phasor plot. I suggest to measure the phasor positions of those two dyes separately and then use the resulting phasor plot positions as centers for the two cursors in the multiplexed image.

-Figure 4:

I don't quite understand the purpose of the experiment shown in Figure 4. The authors attribute the observed changes in lifetime to illumination-induced photosensitization of the dye (Flipper-TR). Do the authors expect any change over time in the membrane tension for biological reasons? If not, why was this example chosen for the demonstration of time lapse imaging with FLIM? Wouldn't it make more sense to select a sample where a change in membrane tension over time is expected based on underlying biological processes? Such process could be externally induced, for example, by the addition of some chemical compound that affects the cell membrane.

Reviewer #3

(Remarks to the Author)

Dunsing-Eichenauer and colleagues developed a technique to perform FLIM measurements in living organoids by combining single-objective light sheet microscopy (soSPIM) with SPAD array detection. The novel part of the approach is the usage of single-objective light sheet microscopy in the scanning and in the static mode. The paper is clear and well written allowing the reader to understand the advantages of their approach over previous confocal alternatives. The example of biological application is limited, though, to one type of sample. In my opinion, the work deserves the publication in Communication Biology if these main and minor points are addressed.

1. You compared the performance of your soSPIM FLIM system with the conventional FLIM implementation on a confocal laser scanning microscope. However, the most direct comparison would be to the existing FLIM light-sheet microscopy systems like the one described by Hirvonen et al., 2020 (<https://doi.org/10.1002/jbio.201960099>). If you cannot run an experiment to directly compare the two systems, could you at least address in the text what are the advantages of your configuration over a TCSPC-based FLIM with light sheet illumination?
2. In the main text, and especially related to figure 1 and 2, you start by describing how your system has the perfect application in the imaging of organoids, but more details related to these organoids are required also in the text and not just in the figure captions. What type of organoids are we talking about? What was labelled?
3. At line 144, you report a different performance of the static from the scanning LS measurements, with a 10% discrepancy probably due to residual pile-up. However, what I think it is missing in the text is a detailed explanation of what are the advantages or in general the differences between the static and the scanning static modality.
4. The description of the JeWell mirrors to circumvent the need of illuminating the sample from multiple sides is very brief. I believe that figure 2b would benefit from the addition of a cartoon where in 3D you show the light paths.
5. One of the limitations of your work is that you show applications related to just one system. Would it be possible to seed other cell types in the JeWells? Like neurons? To generalize the applicability of your system. An experiment similar to Ma et al., 2021 (<https://doi.org/10.1073/pnas.2004176118>) would strengthen the manuscript by providing demonstration of the temporal resolution of your system.
6. In Fig.4, you show the capability of your system to do live volumetric time-lapse membrane tension imaging. There, you also observe a decrease in membrane tension that you explain as a potential photosensitization of the Flipper-TR. Could it reflect a real biological change in membrane tension due to cell division or phototoxicity? Could you prove that it is not phototoxicity by exposing cells to different excitation powers? If it is not related to phototoxicity you should be able to observe the same variation in t2 for slightly higher or lower excitation power?
7. With your system, can you explore any novel biological insight that was not observable with previous technologies? Can you at least comment on that?
8. At line 171, you refer to Flipper-TR without explaining what that actually is, its definition comes later at line 215. Can you move it when you first refer to it?

Version 1:

Reviewer comments:

Reviewer #1

(Remarks to the Author)

Thank you to the authors for considering my comments and revising the manuscript thereafter. Regarding the first two general comments, I would argue that the main advantage of OPM-style approaches compared to soSPIM is their compatibility with standard sample mounting on cover slips or multi-well plates. An aspect not to be underestimated when it comes to disseminating the technology outside highly specialised labs. Just like the authors, I look forward to seeing if/how this SPAD array technology can be incorporated also in such systems.

Furthermore, though there are naturally deeper technical aspects to be had regarding some topics, the authors have addressed my other comments to a satisfactory degree, and I believe the technical detail is sufficient for this type of publication, given the references to previous work.

All in all, I believe this manuscript now represents a satisfactory presentation of the soSPIM-FLIM system, presenting both its potential benefits but also disclosing its limitation. I believe the technology can prove to be a valuable tool by enabling volumetric FLIM imaging in samples compatible with the JeWells.

Andreas Bodén

Reviewer #2

(Remarks to the Author)

The authors have satisfactorily addressed the questions and comments that I had raised during the initial review.

Reviewer #3

(Remarks to the Author)

Dunsing-Eichenauer & Hummert et al. in the revised version of their manuscript titled Fast volumetric fluorescence lifetime imaging of multicellular systems using single-objective light-sheet microscopy, addressed my comments and concerns clearly and rigorously. In particular, the re-analysis of re-Flipper-TR data acquired at different powers is important to state potential effects due to phototoxicity. The quality of the work, already well-written but in need of some clarifications, has now improved significantly. I understand that the application of soSPIM to other biological systems is beyond the scope of this work. For the aforementioned reasons, I think that the manuscript should be accepted for publication in Communications Biology

Response to reviewers

Please find below our point-by-point response to the reviewers' comments.

Reviewers' comments:

Reviewer #1 (Remarks to the Author):

In the manuscript Fast volumetric fluorescence lifetime imaging of multicellular systems using single-objective light-sheet microscopy Dunsing-Eichenauer & Hummert et al. present an approach to light sheet Fluorescence Lifetime Imaging Microscopy utilising a Single Objective Light Sheet (or single objective Selective Plane Illumination Microscopy – soSPIM) configuration. I would firstly like to commend the authors on a well-written and clearly presented scientific work that was enjoyable to read. The manuscript is well-written and easy to understand, and the authors elegantly point out both the strengths and potential limitations of their technology. The authors also present a detailed and quantitative comparison of the performance of their soSPIM approach to specific confocal FLIM versions in terms of imaging speed and signal strength. This specific comparison would benefit from adding a comparison of the axial resolution between the light sheet and confocal approaches.

We thank the reviewer for his positive and constructive feedback. In the revised manuscript, we address the difference in axial resolution and, consequently, sectioning and image quality between confocal and light-sheet imaging modes (e.g. ll. 156 ff., see detailed responses below).

The optical setup that enables light sheet imaging using a single objective is based on a previous publication and cited as such. To enable FLIM imaging, a pulsed 488 nm laser and a 512x512 SPAD array detector are incorporated into the setup. The single objective light sheet configuration allows optically sectioned FLIM images to be acquired at speeds far surpassing that of confocal-based systems. Numerous works have previously explored ways of combining light sheet with FLIM using both very different but also very similar types of detection devices, as stated and cited by the authors. By enabling the creation of the light sheet and detection of the fluorescence through the same objective, the authors remove the need for additional bulky optical elements around the sample and allow for a high-NA detection objective to be used, but instead introduce the requirement of a highly specialised sample mounting chip to reflect the excitation light and create the light sheet. It is also noted that the previous work cited in ref 13 presents an alternative way to use a high-NA detection objective (1.49 NA) in light sheet FLIM. A brief mention of other approaches to single objective light sheet systems such as SCAPE/OPM and their compatibility with FLIM would also be welcome.

In the revised version, we have added an overview table summarizing previous light-sheet FLIM work to the discussion (ll. 312 ff.). We furthermore explicitly mention REF 13 as, to our knowledge, the only previous work with such high detection NA in the introduction (ll. 81 ff.) and discussion (l. 310). Nevertheless, the therein used setup also requires a specialized component, i.e. a tilt illumination device.

We agree that exciting future applications can be realized by implementing FLIM as presented here on SCAPE and OPM light-sheet systems. In this context, state-of-the-art OPM systems operating with high effective detection NA (e.g. Snouty as presented in Sapoznik et al., eLife 2020; DaXi as in Yang et al., Nature Methods 2022) are advantageous over the first SCAPE implementations operating at rather low effective NA. A key advantage of OPM lies in facilitating high-speed volumetric imaging. Time-gated FLIM detection, however, restricts the acquisition speeds that can be achieved. To explore this limit quantitatively, we have sub-sampled the fastest acquisitions presented in this work (static LS FLIM with 100 ms acquisition, i.e. 25 gates with 3 ms exposure per gate) and obtained

robust lifetimes at ca. 65 ms acquisition time (see Fig. S13), with compromises even below. Thus, combining high-speed OPM imaging (which also operates with static illumination) with time-gated detection could allow to image samples at volumetric acquisition rates above 1 VPS, with speeds only limited by the exposure times required for time-gated detection.

We have added the new analysis (Fig. S.13) and an outlook regarding adoption on OPM system to the discussion (ll. 320 ff.).

Overall, the manuscript is clear and presents an approach to light sheet FLIM imaging which has not been previously demonstrated, and the alternative approach to a light sheet FLIM system not requiring multiple objectives is attractive. For certain applications that are compatible with the sample chip, the technology may provide a new and impactful readout, but the wide applicability and adoptability of the method will be restricted by the need for highly specialised sample mounting chips, which make up a critical component of the system.

We agree that soSPIM requires specialized sample-mounting chips, but the simplicity of the optical system make it an affordable and easy to implement alternative to multi-objective SPIM setups. The here utilized mounting chips containing 45° reflective mirrors are compatible with many relevant biological samples including organoids (e.g. mouse and human gastruloids, intestinal, hepato- and neuroectoderm organoids), spheroids (e.g. oncospheres), model organisms (*Drosophila*, *C.elegans*) and culture cells (e.g. U2OS, S180) (Galland et al. 2015, Singh et al., 2017, Beghin et al., 2022, Marchand et al., 2025), which we mention now in the introduction (ll. 74 ff.).

Nevertheless, we also aimed to demonstrate the great advantage of the here utilized gated SPAD array detector for light-sheet based FLIM, providing much higher count rates than previously employed time-resolved cameras (see Table 1, ll. 312 ff.). This technology is compatible with many light-sheet microscopy systems, provided a high collection efficiency can be achieved. We have added a paragraph to the discussion to emphasize this aspect (ll. 320 ff., see also our reply to the previous comment).

Below are more specific comments on certain points in the manuscript that I hope can help strengthen the work and clarify some points.

In Fig 1, the highest 0.1% of the pixels are excluded from the quantification. A clearer explanation for this would be good.

In our quantification, we aim to have a metric for the brightest pixels in the image that is robust with respect to outliers and hot pixels. Accordingly, the pixels at 99.9% counts corresponds to a representative bright pixel but not an outlier. We have modified the caption accordingly.

The sentence starting with “In contrast” on line 148 does not clearly describe what I believe it is intended to say. I would suggest (if I’ve understood it correctly) re-writing the sentence to more clearly explain that it is the degradation of image quality along the light sheet propagation direction that is discussed.

We thank the reviewer for his suggestion to discuss degradation artefacts in confocal and light-sheet FLIM in more detail. We have added a supplementary figure (Fig. S8) showing a quantitative comparison of the image quality of confocal and light-sheet acquisitions and added a respective paragraph to the results section (ll. 156 ff.), rephrasing and extending the original statement.

In Fig 2, the above-mentioned image degradation is supposedly shown, but the figure (a) does not indicate where the light sheet illumination is coming from.

We have modified Fig. 2a to indicate where the light-sheet is coming from and rotated images so that the illumination comes from the left side for both scanning and static mode. In addition, we have added a supplementary figure (Fig. S8) which shows quantitatively the image degradation in the propagation direction of the light-sheet and the improvement obtained with sequential dual-sided illumination and subsequent image fusion.

Relating to Fig 2, but also to the manuscript as a whole, I would invite a more in-depth discussion on the potential benefits of the scanned light sheet. As it stands, very few benefits are presented, and it is not entirely clear why it has been included as an alternative method at all.

We thank the reviewer for pointing this out. We have added two sentences to the discussion which emphasize the advantages of scanned over static light-sheet illumination, namely 1) improved image quality in thick, heterogeneous specimen (evident from higher DCTS score in the newly added Fig. S8) due to more homogenous illumination and less artifacts by scattering, resulting in reduced shadowing and blur (l. 301, l. 304), 2) compatibility with more advanced illumination schemes, e.g. two-photon excitation and Bessel beams (ll. 332 ff.).

In the example application, measurements are taken on an organoid expressing H2B-GFP and stained with Flipper-TR. The reference measurements for the individual lifetimes are also taken in an organoid containing both labels. Why not take the reference measurements for the different labels on organoids containing only one or the other label?

We agree that reference measurements in single-species samples can be used to determine reference lifetimes. However, we here wanted to show that, provided sufficient spatial segregation, i.e. in the presence of “pure” pixels with signal stemming predominantly from one species, these additional measurements are not needed and cursor selection in the phasor plot can be used to determine individual lifetime patterns directly in mixed samples. The feasibility of this procedure is not only evident from the clear separation of nuclei and membrane pixels, but also from the quantitative agreement of the lifetimes extracted from the pixel populations selected in the phasor plot with the confocal lifetimes determined by manual annotation of membrane and nuclei masks.

We have added a comment emphasizing that individual reference measurements need to be performed for strongly overlapping structures (ll. 206 ff.).

In addition to the above, the Flipper-TR is a mechanosensitive label and, at first thought, does not seem like the most suitable label to use in the multiplexing experiment due to the potentially varying lifetime. A more comprehensive discussion as to why this was chosen would help.

We agree with the reviewer that the Flipper-TR probe is not ideal for demonstrating lifetime multiplexing. It nevertheless provides a suitable label here for the following reasons:

- 1) Measurements were acquired in 48h gastruloids, i.e. at a stage where the cellular composition of gastruloids is still very homogenous as mESC cells in the aggregate have not differentiated yet. Confocal FLIM of Flipper-TR at this stage has indicated no strong heterogeneities (see Box1 in Roffay et al., Nature Protocols 2024)
- 2) Lifetime differences reported by Flipper-TR corresponding to potentially present variations in membrane tension are typically on the order of few hundred ps at maximum (often 0.1-0.2 ns, see Roffay et al., Nature Protocols 2024), which is small compared to the lifetime difference of Flipper-TR and GFP (intensity weighted difference of 1.2 ns),
- 3) Generating endogenous mESC lines is tedious and existing lines typically contain spectrally non-overlapping fluorophores. Thus, labeling live organoids with spectrally overlapping fluorophores exhibiting different lifetimes and targeting structures that can be easily discriminated visibly (like nuclei and membranes) is not trivial.

We have added a sentence to the multiplexing results section to explain this choice of labels (ll. 191 ff.).

In Fig 4, there seems to be quite a large variation in the initial τ_2 lifetimes measured over the three different spheroids. This difference is not clearly explained in the text or figure legend. Could you please comment on this? Is this due to pile-up effects?

Residual pile-up could be a cause of the observed variation, but we do not observe a clear correlation with counts within organoids. In addition, lifetimes do not scale with illumination power and are fairly stable with depth across organoids (see Fig. S11). Since the data were acquired at different illumination powers, illumination induced effects could also play a role. Lastly, we cannot exclude an actual biological heterogeneity, e.g. due to the presence of a lumen in organoid 3 or organoid size. We have added a comment in the results mentioning the potential factors involved (ll. 247 ff.).

In methods, the light sheet waist is stated to be 1.5/1.9 μm with a Rayleigh length of 50/80 μm . These numbers would require some clarification as to their exact meaning, the waist seems thinner than expected from my calculations.

We thank the reviewer for this comment and apologize that there was indeed a mistake in the estimation of the light-sheet dimensions. We have amended this and now clearly define the given values (ll. 378 f.). We have also re-calculated the estimated illumination power densities (l. 117, Table S1).

In the legend of figure S2 an intensity loss of 25-30% is mentioned. However, it is very hard to see or understand what these percentages refer to or how they are calculated.

The indicated loss referred to the reduced signal detected in confocal FLIM inside organoids at depth compared to outer cell layers. We have now added a separate figure (Fig. S8) on image degradation and quantitative comparison of image quality between confocal and light-sheet acquisitions, including a graph of intensity vs. distance to the edge of organoids for confocal data (Fig. S8b), referred to in l. 161 of the results.

Reviewer #2 (Remarks to the Author):

In their manuscript, Dunsing-Eichenauer et al. used a SPAD array detector to implement fluorescence lifetime imaging with 10-100 times faster acquisition times compared to confocal FLIM. The high imaging speed is a result of the parallel probing of image pixel as compared to the sequential, pixel-by-pixel acquisition with a laser scanning microscope. To demonstrate the capability of their setup, the authors imaged organoids expressing H2B-GFP and stained with Flipper-TR. Using phasor FLIM analysis, the two dye species could be separated from each other. Further, SPIM-FLIM was used for volumetric time-lapse imaging of membrane tension in organoids.

Overall, the data presented seems solid and scientifically sound. High speed FLIM data acquisition with optical sectioning is highly desired for biological studies and biomedical imaging. The manuscript is well written and the experiments are described with sufficient detail. In my opinion, the manuscript is well deserving of publication in Communications Biology after addressing the following comments and questions:

We thank the reviewer for his positive feedback and address the questions below.

-Figure 2a/Lines 143-144:

There is a 10% systematic deviation between the average pixel lifetime determined from static light-sheet measurements compared to scanning light-sheet measurements. The authors claim that this difference may be a result of the residual pile-up in scanning light-sheet in high intensity pixels. To test this hypothesis, I suggest to perform an analysis with those high intensity pixels excluded from the data.

We thank the reviewer for this idea and have performed the suggested analysis, presented in Fig. S9. The data support our hypothesis that residual pile-up is responsible for the slight overestimation of lifetimes with scanning illumination, since the difference to the confocal reference drops from 5.2% (overestimation) when including all pixels to 2.5% when excluding the pixels above median intensity and is negligible when excluding all pixels above the bottom quartile. We state this in ll. 153 ff. of the results.

-Figure 3a:

Two cursors were used to select lifetime pixel on the phasor plot representative of H2B-GFP and Flipper-TR. While the selection makes sense based on the resulting masked images, it would be better to select the position of those cursors based on the expected/measured positions of the lifetimes of H2B-GFP and Flipper-TR on the phasor plot. I suggest to measure the phasor positions of those two dyes separately and then use the resulting phasor plot positions as centers for the two cursors in the multiplexed image.

We agree that determining the phasor positions in individual reference samples would provide an alternative strategy and would be favorable for spatially overlapping structures. We refer to our reply to the same question raised by reviewer 1. We here aimed to show that separate reference measurements are not needed, and that the phasor plot can be used to generate masks of pure pixels and determine species lifetimes from the corresponding decays which are afterwards used for a fitting-based decomposition of the entire image. The determined species lifetimes agree very well with the confocal lifetimes determined by manual annotation of membrane and nuclei masks, validating this procedure. In addition, the obtained Flipper-TR lifetimes agree well with the values measured on gastruloids stained only with Flipper-TR (see Fig. 4). Note also that the phasor plot is not used for decomposition, only for identifying and masking “pure” pixels.

We have added a comment emphasizing that individual reference measurements need to be performed for strongly overlapping structures (ll. 206 ff.).

-Figure 4:

I don't quite understand the purpose of the experiment shown in Figure 4. The authors attribute the observed changes in lifetime to illumination-induced photosensitization of the dye (Flipper-TR). Do the authors expect any change over time in the membrane tension for biological reasons? If not, why was this example chosen for the demonstration of time lapse imaging with FLIM? Wouldn't it make more sense to select a sample where a change in membrane tension over time is expected based on underlying biological processes? Such process could be externally induced, for example, by the addition of some chemical compound that affects the cell membrane.

We agree with the reviewer that a biologically controlled experiment with external perturbation would be the ideal proof-of-concept experiment. We here aimed to perform, first, a more technical proof-of-concept experiment for 3D time-lapse FLIM imaging with a biologically relevant probe, demonstrating the advantage of the here achieved, unprecedented volumetric speed and photon throughput compared to conventional FLIM for time-lapse measurements of living specimen. We have now performed additional analyses of the Flipper-TR data, further substantiating the observation of illumination induced lifetime changes of the Flipper-TR probe (see updated Fig. 4, related text and our reply to the related question of reviewer 3). Such changes have not been reported yet previously, because available FLIM instrumentation has so far not allowed imaging live specimens at this

resolution in 3D over prolonged periods of time. Notably, we were able to detect the reported lifetime changes not only when averaging over entire organoids, but even in individual cells tracked in 3D. This represents a significant advancement, as it has not been possible before to extract meaningful lifetimes in such short acquisition times from individual living cells. Nevertheless, the observed illumination induced effects make controlled applications more difficult, as lifetime changes induced by illuminating the probe and by externally induced changes in membrane tension (e.g. osmotic shock) would superimpose. Therefore, future work is required to apply time-lapse 3D FLIM imaging to other probes in controlled biological settings, which we state in the discussion (ll. 341 ff.), but this goes beyond the scope of this work.

Reviewer #3 (Remarks to the Author):

Dunsing-Eichenauer and colleagues developed a technique to perform FLIM measurements in living organoids by combining single-objective light sheet microscopy (soSPIM) with SPAD array detection. The novel part of the approach is the usage of single-objective light sheet microscopy in the scanning and in the static mode. The paper is clear and well written allowing the reader to understand the advantages of their approach over previous confocal alternatives. The example of biological application is limited, though, to one type of sample.

In my opinion, the work deserves the publication in *Communication Biology* if these main and minor points are addressed.

1. You compared the performance of your soSPIM FLIM system with the conventional FLIM implementation on a confocal laser scanning microscope. However, the most direct comparison would be to the existing FLIM light-sheet microscopy systems like the one described by Hirvonen et al., 2020 (<https://doi.org/10.1002/jbio.201960099>). If you cannot run an experiment to directly compare the two systems, could you at least address in the text what are the advantages of your configuration over a TCSPC-based FLIM with light sheet illumination?

We thank the reviewer for this comment. To compare the results of our work to the FLIM performance of existing FLIM light-sheet systems, we have added a table to the discussion (see table 1, ll. 312 ff.). The Setup by Hirvonen et al. possessed a much lower detection NA (0.5) and throughput (e.g. <100 kHz overall count rate compared to ca. 40 MHz/ 400 MHz achieved in our work with scanning/ static light-sheet illumination). The much higher photon yield results in a substantial speed advantage (e.g. 5 s fastest frame time in Hirvonen et al. compared to 0.6 s/0.1 s in our work). To highlight the high pixel and overall count rates achieved in our light-sheet FLIM implementation, we have added a panel in Fig. 1 showing the count rates of all acquisitions.

Overall, our implementation provides superior speed, collection efficiency and photon throughput compared to all previous light-sheet FLIM systems, now stated in ll. 307 ff. of the discussion.

2. In the main text, and especially related to figure 1 and 2, you start by describing how your system has the perfect application in the imaging of organoids, but more details related to these organoids are required also in the text and not just in the figure captions. What type of organoids are we talking about? What was labelled?

In this work, we have used mouse gastruloids, e.g. aggregates of mouse embryonic stem cells mimicking stages around gastrulation in the mouse embryo. Gastruloids are not only a relevant biological model system, but also represent key challenges for 3D imaging as they are highly opaque and scattering samples (see Gros et al., *eLife* 2025). We have added a short description of gastruloids in the beginning of the results section (ll. 96 ff.). All labels are given in the results text describing each experiment (e.g. fixed gastruloids in which E-cadherin was immunostained with AF488 for the

benchmark, live gastruloids expressing endogenous H2B-GFP and labelled with Flipper-TR for multiplexing, live gastruloids stained with Flipper-TR for time-lapse 3D FLIM imaging).

3. At line 144, you report a different performance of the static from the scanning LS measurements, with a 10% discrepancy probably due to residual pile-up. However, what I think it is missing in the text is a detailed explanation of what are the advantages or in general the differences between the static and the scanning static modality.

Regarding the small difference in lifetimes detected with scanning and static illumination, we have performed an additional analysis showing that lifetimes detected with scanning LS at low to moderate count rates match very well with the ones detected with static LS (Fig. S9, ll. 153ff.).

Moreover, we have added a quantitative analysis of image quality in scanning and static LS measurements (Fig. S8), revealing a trade-off between FLIM performance with gated detection (which is superior with static LS due to higher count rates and less pile-up) and image quality (which is superior with scanning detection), see ll. 297 ff. added to the discussion. For this reason, scanning LS is the preferred modality for larger 3D specimen. Additionally, it allows implementing more advanced illumination modes (such as Bessel beams and two-photon light-sheet), for which FLIM as implemented here can be adopted (now stated in the discussion ll. 332 ff.).

4. The description of the JeWell mirrors to circumvent the need of illuminating the sample from multiple sides is very brief. I believe that figure 2b would benefit from the addition of a cartoon where in 3D you show the light paths.

We agree with the reviewer that the description may have been confusing. We have added a sketch to Fig. 2b illustrating dual-sided illumination using the JeWells and also rotated all images in Fig. 2a so that the illumination comes from the left side. The advantage of the JeWell mirrors is not to circumvent illuminating the sample from multiple sides, but to provide four illumination sides without the need of four illumination arms and objectives.

5. One of the limitations of your work is that you show applications related to just one system. Would it be possible to seed other cell types in the JeWells? Like neurons? To generalize the applicability of your system. An experiment similar to Ma et al., 2021 (<https://doi.org/10.1073/pnas.2004176118>) would strengthen the manuscript by providing demonstration of the temporal resolution of your system.

Applications of high-resolution soSPIM imaging using JeWells have been demonstrated for numerous biological samples, including 2D cell culture (e.g. U2OS, S180), model organisms (e.g. *Drosophila*, *C.elegans*), organoid samples (e.g. mouse and human gastruloids, intestinal, hepato- and neuroectoderm organoids) and spheroids (e.g. oncospheres) (Galland et al. 2015, Singh et al., 2017, Beghin et al., 2022, Marchand et al., 2025), which we now mention in the introduction (ll. 74 ff.). We agree that applications in other biological systems would be exciting, but establishing an additional biological system goes beyond the scope of this work.

To address the temporal resolution of our system and speed limit that can be achieved, we have computationally sub-sampled static LS soSPIM-FLIM images acquired at 100 ms frame time (25 gates, 3 ms exposure per gate) and computed lifetimes with fewer gates (Fig. S13). Robust lifetimes can be obtained with acquisition times of ca. 65 ms (i.e. 15 gates of 3 ms exposure each), with compromises even below. Thus, depending on the precision needed, a time resolution of ca. 50 ms can be achieved with our system. We now address this in the discussion in the context of high-speed OPM imaging systems (ll. 320 ff.).

6. In Fig.4, you show the capability of your system to do live volumetric time-lapse membrane tension imaging. There, you also observe a decrease in membrane tension that you explain as a potential

photosensitization of the Flipper-TR. Could it reflect a real biological change in membrane tension due to cell division or phototoxicity? Could you prove that it is not phototoxicity by exposing cells to different excitation powers? If it is not related to phototoxicity you should be able to observe the same variation in t_2 for slightly higher or lower excitation power?

We thank the reviewer for this comment and have re-analyzed Flipper-TR data acquired in different organoids at different illumination powers (see novel panels in Fig. 4). We observe the same overall trend of lifetimes and intensity in both organoids (initial decrease in lifetime and intensity and subsequent recovery of the intensity). However, in the organoid illuminated with higher power (organoid # 3), the lifetime change occurred much faster than in an organoid illuminated with lower excitation power (# 2). While we had previously only analyzed lifetimes in individual cells, we now observe the same trend when averaging over the entire organoid. Given that mESCs only divide ca. every 10 h and differentiate on the scale of hours, it is unlikely that this reflects membrane tension changes due to specific biological events, e.g. divisions or differentiation. Rather, the fact that the rate of the changes is power dependent strongly indicates an illumination induced process. At this point, we cannot discriminate between changes of membrane composition/ tension due to potential photosensitization and phototoxicity, which may be related. We have therefore modified the discussion to mention phototoxicity as a potential mechanism (ll. 257 ff.).

We have modified the results section and figure 4 to include the novel analyses.

7. With your system, can you explore any novel biological insight that was not observable with previous technologies? Can you at least comment on that?

The unprecedented speed of our FLIM approach opens the door to many biological applications in the context of larger 3D specimen. In particular, we anticipate applications in the context of high-throughput screening of complex, dynamic 3D tissues such as organoids, e.g. using FLIM based biosensors, 3D live mapping of molecular interactions or forces via FLIM-FRET, and 3D live metabolic imaging (e.g. gradients of oxygenation in organoids or embryos via FLIM based oxygen sensors), which are currently only accessible in 2D due to speed limitations. In addition, combining FLIM detection as presented here with high-speed volumetric light-sheet systems (e.g. OPM) may facilitate studies of fast dynamics (e.g. migratory cells) and subcellular processes (e.g. pH changes in subcellular compartments) at volumetric rates above 1VPS, which goes far beyond currently accessible time scales of confocal FLIM imaging. We have added these aspects to the discussion (ll. 329 ff., ll. 346 ff.).

8. At line 171, you refer to Flipper-TR without explaining what that actually is, its definition comes later at line 215. Can you move it when you first refer to it?

Amended. See also our reply to the related comment raised by reviewer 1.